# Study on the Influence Mechanism and Space Distribution Characteristics of Rail Transit Station Area Accessibility Based on MGWR

**DOI:** 10.3390/ijerph20021535

**Published:** 2023-01-14

**Authors:** Daoyong Li, Hengyi Zang, Demiao Yu, Qilin He, Xiaoran Huang

**Affiliations:** 1School of Architecture and Art, North China University of Technology, Beijing 100144, China; 2Centre for Design Innovation, Swinburne University of Technology, Hawthorn, VIC 3122, Australia

**Keywords:** rail transit, accessibility, MGWR, spatial heterogeneity, internet map

## Abstract

The accessibility of rail transit station areas is an important factor affecting the efficiency of rail transit. Taking the Beijing rail transit station area as our research object, this paper took a 15 min walking distance as the index of station area accessibility, and investigated the status quo and influencing factors of the unbalanced distribution of rail transit station area accessibility in Beijing. In this paper, the data of Beijing rail transit stations were obtained from the Amap open platform, and the accessibility of the station area was calculated using the path planning service provided by the Amap API. The Getis–Ord Gi* method was used to analyze the overall distribution characteristics of the accessibility of the Beijing rail transit station area, then the high accessibility area and the low accessibility area were determined. To explore the factors influencing domain accessibility, multi-source data were obtained, a total of 11 indicators were constructed, and the random forest model was used to identify feature importance. Using the eight selected influencing factors, the OLS regression model, GWR model, and MGWR model were used to study the spatial heterogeneity of influencing factors. By comparison, it was concluded that the MGWR model can not only effectively analyze the spatial heterogeneity of rail transit station accessibility, which can automatically mediate the bandwidth of different influencing factors, and then reflect the spatial changes of the influencing factors of rail transit station accessibility more truly. The results show that (1) the accessibility of the Beijing rail transit station area shows obvious spatial agglomeration characteristics in space. The accessibility of the station area in the fourth ring is higher than that outside of the fourth ring road, and the accessibility near the south and north fifth ring road is higher than that of the east fifth ring road and the west fifth ring road. (2) The basic influencing factors of rail transit station accessibility include road density and functional mixing degree.

## 1. Introduction

In recent years, rail transit has been in a period of rapid development in China, which has become an important mode of travel for urban residents, and its impact on urban development is mainly reflected in the range of stations [1]. In addition, as an economic center with a population of more than 20 million, the development core gradually shifts to the edge as the population continues to grow [2]. Rail transit can provide greater accessibility for human activities, support sustainable transport patterns, and guide the development of the areas along the route [3]. As a public transport system supporting the sustainable development of cities, rail transit station areas are also of great significance to the development of urban areas [4]. At the same time, rail transit station area accessibility, as a spatial perception of users, is the result of the combined effects of various factors in the station area, and also reflects the satisfaction of the last kilometer of public transit travel [5]. At present, rail transit station area accessibility has become an important research direction in the field of rail transit [6]. Improving the accessibility of rail transit station areas is an important measure to improve travel satisfaction. Exploring the distribution status and influencing factors of accessibility can provide a basis and guidance for improving station area accessibility.

The study of accessibility was first applied to public service facilities [7]. In 1959, Hansen defined accessibility for the first time to indicate the difficulty in reaching the destination [8], which was subsequently widely applied to urban and rural planning, geographic mapping, and other disciplines. The research on rail transit accessibility is mainly divided into the research on rail transit station area accessibility and the research on rail transit network accessibility. For example, Cheng Changxiu et al. studied the accessibility of subway lines, transfer stations, and terminal stations in Beijing [9]. Wei Panyi et al. studied Beijing subway lines based on bus transfer and accessibility between rail stations [10]. Yao Zhidang et al. optimized the linear buffer method and proposed the road network distance buffer method and the road network attenuation method [11]. Ma Shuhong et al. calculated the accessibility of different travel modes and station areas by determining the radiation range of different connection modes of stations [12].

There is a wide range of studies focusing on the spatial correlation between cities (urban context) and railway stations, which cannot be ignored, and with insights into accessibility that need to be implemented, especially in the European context, which has been thinking about these issues for some time [13,14,15,16,17,18]. Zuo et al. [19] studied the relationship between the slow road network connectivity of bus stations and bus travel. Zuo et al. [20] also discussed the influence of slow traffic on the accessibility of public transport, but the object was the whole process of bus travel. Although Wei Pan et al. studied the accessibility of station areas, they mainly focused on the convenience of bus transfer and transfer between stations [10]. Kusumo found in a comparative study of urban accessibility: a well-integrated street network can make rail stations more accessible [21].

Based on the above, we found that past studies are limited to the following aspects: Some studies on rail transit accessibility focus on the level of the station area, which reflects the service level of the last kilometer of rail transit travel. Carra, M et al. consider station area accessibility by applying a backtracking algorithm that optimizes distances by determining the pedestrian path with the shortest distance [22]. The research content also lacks research on the spatial distribution of station accessibility, does not pay attention to the reasons for the unbalanced distribution of accessibility, and lacks discussion on its influence mechanism. Moreover, GWR and MGWR are rarely applied to the study of spatial characteristics of rail transit station area accessibility and the interpretation of related factors.

In addition, from the perspective of station accessibility characteristics, due to the similarity of spatial factors, the station spaces with close distances are generally closer to each other in terms of station accessibility. The spatial distribution of the accessibility of rail transit stations is neither scattered nor random, but shows the characteristics of clustered distribution. Different accessibility distributions are formed in the urban rail transit station space, indicating that the accessibility of the station area has certain spatial differentiation characteristics, and the influence of different influencing factors on the accessibility of the station area is also different in space.

Previous studies on the influencing factors of accessibility are also mostly used as homogeneity hypothesis analysis. The spatial difference between the impact of built environment and social environment on accessibility is not considered. Therefore, the random forest algorithm, OLS, GWR, and MGWR models are used to analyze the spatial distribution characteristics and influencing factors of station accessibility, to fill the above gaps. In order to better understand the influence mechanism of station accessibility, this paper provides a reference for rail transit station areas.

The remainder of the paper is as follows: Section 2 provides an overview of the study areas and data sources. Section 3 describes the method of index screening and introduces the principle of the MGWR model. Section 4 explains the findings and discussion. Section 5 focuses on the significance and limitations of this study.

## 2. Study Area and Data Sources

### 2.1. Study Area

Beijing is the capital of China and is located in the north of the North China Plain, adjacent to Hebei province and Tianjin City. Beijing rail transit has a history of nearly 60 years, currently including 783 km of urban rail transit and 365 km of urban suburban railway. According to the Beijing Rail Transit Network Plan (2020–2035), Beijing’s urban rail transit will reach 1625 km, and the total scale of the rail transit network will reach 2683 km [23].

Therefore, this paper takes Beijing as the main research area, and the research objects not only include the rail transit stations in Beijing administrative region, but also include some stations in Langfang, Hebei Province, which are extended by Beijing rail transit. This paper mainly studies the influencing factors and spatial distribution characteristics of the rail transit station area accessibility in the region. The research area is shown in Figure 1.

### 2.2. Data Sources

#### 2.2.1. Rail Transit Station Location Data

This paper takes Beijing rail transit stations as the research object. We use Python from Amap open platform (https://lbs.amap.com/api/webservice/guide/api/search, accessed on 14 March 2021) in the search area, the city of Beijing subway stations as keywords. The results returned by Amap POI were obtained, and the location data of 436 rail transit stations were obtained, among which 4 stations were located in Sanhe Country Langfang City, Hebei Province.

#### 2.2.2. Rail Transit Station Area Data

The Amap path planning service was used to obtain the path and time of travel in different ways at specified two points. According to the Urban Master Plan of Beijing (2016–2035) [24], the time of life circle is set as the time cost of 15 min walking. Therefore, this paper adopts the 15-minute walking range of the station as the rail transit station area. The specific scope is obtained by the path planning service of the Amap Open Platform, and the specific method will be explained in Section 3.2.1.

#### 2.2.3. Data of Influencing Factors

The data used to calculate the influencing factors of the rail transit station area accessibility are from many aspects. Including the administrative border data from the Aliyun big data platform (http://datav.aliyun.com/portal/school/atlas/area_selector, accessed on 3 June 2022); POI data and road network data from Amap Open Platform (Alibaba, China) (https://lbs.amap.com/api/webservice/guide/api/search, accessed on 14 July 2022); 2020 population distribution data from WorldPOP (https://www.worldpop.org, accessed on 16 July 2022); Housing estate price data from Lianjia.com (https://cm.lianjia.com, accessed on 13 August 2022); Building-land data from GlobleLand30 (http://globeland30.org/defaults.html?src=/Scripts/map/defaults/download.html&head=download&type=data, accessed on 20 August 2022). The data description is shown in Table 1.

## 3. Method

### 3.1. Research Framework

Based on the actual situation, this study takes the walkable range of rail transit stations as the basic research unit to study the distribution of rail transit station accessibility in Beijing and explore the factors that affect the distribution of rail transit station area accessibility. This study can provide a basis and guidance for improving station area accessibility.

Firstly, the station area range and accessibility are evaluated, and their spatial distribution characteristics are analyzed. Secondly, the indicators are screened, and finally, the OLS model, GWR model, and MGWR model are compared and analyzed to interpret the influence mechanism of station reachability. We obtained the data of Beijing rail transit stations from the Amap open platform, and used the path planning service provided by the API of Amap Road to calculate the 15 min walking accessibility range of all stations as the station area accessibility. The Getis–Ord Gi* method was used to analyze whether there was spatial heterogeneity in the influencing factors of station accessibility, and the spatial distribution of station area accessibility to determine the areas with high and low accessibility for future optimization of the station area. In order to explore the influencing factors of the distribution of station accessibility, this study obtains multi-source data including POI data, remote sensing data, road network data, and demographic data, and constructs 11 indicators. Finally, random forest regression was used to screen the influencing factors, and eight influencing factors and rail transit station accessibility were selected to construct the least square method (OLS) model, GWR model, and MGWR model, respectively. Through the analysis results of the models, the influencing factors of the current distribution of rail transit station accessibility were explained so as to provide guidance to urban planners and other decision makers. This study provides a new idea for studying the spatial utilization pattern and influencing mechanism of rail transit station areas (Figure 2).

### 3.2. Evaluation of the Accessibility Distribution Characteristics of the Rail Transit Station Area

#### 3.2.1. Acquisition and Accessibility Assessment of the Rail Transit Station Area

Accessibility measurement methods usually include the nearest distance method, gravity model method, and coverage method [25]. There are also several types of accessibility, such as utility-based, individual-based, and distance-based [26]. Among them, the coverage method is based on the scope of services as a standard, which is more intuitive than other methods and more convenient to explore the relationship between accessibility and the spatial differentiation characteristics of influencing factors. Distance- and time-based accessibility is chosen to better reflect the rail station area and reachability. Calculating reachability with GIS platforms requires a lot of data work and modeling work [27]. This paper calculates the corresponding accessibility evaluation index based on the platform and data of the Internet map provider, and its calculation method can eliminate the cumbersome data collection and modeling process, and is also more in line with the practical life scene [28]. In this paper, the 15-minute walking range of rail transit stations is taken as the station range and accessibility, and the route planning service provided by Amap open platform (https://lbs.amap.com/api/webservice/guide/api/direction, accessed on 20 September 2022). The walking speed is calculated by the Amap platform based on big data [28]. In order to prevent the influence of special factors on the result of path planning, the relevant data were crawled at 20:15 on 19 September 2022.

Amap path planning service can provide different routes and times of different travel modes between two points. The calculation not only can take into account the ups and downs of the terrain, but also the influence of traffic conditions. So, the calculation of travel time is more consistent with daily life.

Taking the current status of rail transit stations in Beijing as an example, the calculated time cost is 15 min, and the range within a 15 min walk around each station is analyzed, respectively. The specific range and accessibility acquisition process of rail transit stations are as follows:Through the POI query service provided by Amap open platform, Python is used to obtain the location information of Beijing rail transit stations in batches, including the station name and specific coordinates, and the coordinate data are processed for easy research and use.A random generator is used to establish a dot array around each site with a radius of 1500 m, and the coordinates of all the points of the lattice are obtained.With the site as the starting point and each point in the dot matrix as the end point, the path time and walking action are planned through the Amap path planning service to obtain the return value.The points with a time of less than 900 s are screened, and the outermost points are connected into a polygon, that is, the 15 min walking range and the station domain accessibility of the site.

In this study, the coverage method was used to calculate the station accessibility of Beijing rail transit, and the data were visualized in ArcGIS (ESRI, Redlands, CA, USA) to obtain the 15 min walking coverage area from the site as the station range and station accessibility.

#### 3.2.2. Cold–Hot Spot Analysis

In order to analyze the spatial distribution characteristics of rail transit station accessibility, this paper uses the spatial auto-correlation tool to study the resource distribution. Cold and hot spots analysis can classify variables into cold spots and hot spots according to the degree of spatial distribution and aggregation, which can preliminarily determine whether variables have spatial clustering distribution characteristics, and can well reflect the cold and hot spots distribution of variables in local spatial areas [29]. The method is as follows:(1)$$G_{i}^{*}=\frac{\sum_{i=1}^{n}W_{i,j}-\bar{X}\sum_{j}^{n}w_{i,j}}{S\sqrt[{2}]{\frac{[n\sum_{j}^{n}w_{i,j}^{2}-{\left(\sum_{j=1}^{n}W_{i,j}\right)}^{2}]}{n-1}}}$$
(2)$$\bar{X}=\frac{\sum_{i=1}^{n}x_{j}}{n}$$
(3)$$S=\sqrt{\frac{\sum_{j}^{n}x_{j}^{2}}{n}-{\left(\bar{X}\right)}^{2}}$$
where *x_i_* and *x_j_* are the attribute value of the features *i* and *j*; *w_i_* and *w_j_* are the spatial weights between feature *i* and feature *j*; *n* is the number of features in the data set. When the statistic of $G_{i}^{*}$ of an element is higher than that of mathematics and passes the hypothesis test, it is a hot spot, and otherwise, it is a cold spot.

#### 3.2.3. Spatial Autocorrelation Analysis

Spatial autocorrelation represents the interaction of multiple variables at different spatial locations [30]. The spatial dependence between the variables can be detected, and its indicators indicate the digital relationship between the observation object and a certain feature in the spatially adjacent area. Due to the influence of spatial location, multiple variables no longer exist independently, presenting a related and random spatial distribution pattern [31]. According to the spatial autocorrelation, we can judge that the analysis of the research object spatial autocorrelation can be divided into global correlation and local correlation, in which the global correlation is the Moran’s (Moran index), which can represent whether the spatial features cluster or outliers at the global spatial scale. Local correlation is called LISA (local indicators of spatial association), which is the correlation analysis of spatial features and their adjacent regions, which can reflect the agglomeration degree of spatial features in the local area more than the global correlation.

### 3.3. Spatial Heterogeneity and Influencing Factors Analysis

#### 3.3.1. Selection of Influencing Factors

In addition to the study of spatial heterogeneity of rail transit station area accessibility, this study also aims to further explore the construction of the influencing mechanism. The influence of surrounding conditions on accessibility is mainly considered, and site characteristics such as the architectural typology of the station are not further analyzed in this study. According to previous studies, the accessibility of rail transit station areas is related to a series of factors, including road density, population density, development degree, land-use nature, etc. [32,33].

The road density is selected from the pedestrian path marked by the Amap. The functional density calculation was carried out by using the POI data to obtain the overall development of the station area, and then the classification density calculation was carried out to show the difference in the main functions of the surrounding space of different station areas.

Referring to similar studies, this study selected the built environment and social environment as the first-level indicators, and a total of 11 factors as the second-level indicators, which were analyzed separately. In order to reduce the error in the analysis, all indexes are normalized (Table 2). The top 8 indexes with significant impact on the accessibility of rail transit stations were selected by random forest regression.

#### 3.3.2. Random Forest Regression Analysis

Random forest is an integrated machine learning algorithm that can explain the role of multiple independent variables on the dependent variable, and can better explain the relationship between the variables [34,35]. Random forest is a Bootstrap resampling method to build a classification tree by random selection of multiple independent variables, and the number of selected independent variables selected is less than or equal to the number of all independent variables. Some of the simultaneous independent variables were selected for determining the categorical tree nodes. Multiple different classification trees were constructed as above to form random forests, and finally, the most repetitive tree from the forest was selected as the final result. In this study, a random forest regression analysis between the above-influencing factors and the TRC station domain accessibility was performed using SPSS Modler 18.0 (IBM, Armonk, NY, USA), to reveal the overall influence of different influencing factors on the accessibility of rail transit stations.

#### 3.3.3. Correlation Analysis

Different influencing factors need to be compared and analyzed in the study. Pearson correlation coefficient is used to classify 11 influencing factors affecting the accessibility of rail transit stations and quantify the correlation between different influencing factors. The specific calculation formula is as follows:(4)$$\rho_{x,y}=\frac{\sum^{\;}\left(X-\bar{X}\right)\left(Y-\bar{Y}\right)}{\sqrt{\Sigma {\left(X-\bar{X}\right)}^{2}+\Sigma {\left(Y-\bar{Y}\right)}^{2}}}$$
where *X* and *Y* are the variable values of the two variables, respectively:$\;\rho_{x,y}$ is the correlation coefficient of the sample, and its value is between [−1,1]. When the $\rho_{x,y}$ value is closer to 0, it indicates that the correlation between the two variables is lower. When the value is closer to 1, it indicates that the two variables show a strong positive correlation. The closer the current value is to −1, it indicates that the two variables show a strong negative correlation.

#### 3.3.4. Multi-Scale Geographic-Weighted Regression (MGWR)

Traditional GWR does not consider the different spatial influence degrees of different influence factors on the research object, and adopts unified bandwidth processing. However, the MGWR uses its own influence degree for each impact factor, which solves the problem that different variables have different influence degrees. In this method, the factors with stable global scale are analyzed by using larger bandwidth, and the factors with strong local correlation are analyzed by using smaller bandwidth. The formula of MGWR is:(5)$$y_{i}=\overset{k}{\underset{\dot{J}=1}{\sum }}\beta_{bwj}\left(u_{i},\nu_{i}\right)x_{ij}+\varepsilon_{i}$$
where $x_{ij}$ is the *j*-th predictor variable.$\;\left(u_{i},\nu_{i}\right)$ represents the coordinates of each site.$\;\beta_{bwj}$ represents the j-th variable regression coefficient. In this study, MGWR 2.2 (Arizona State University, Tempe, AZ, USA) was used for the regression analysis of the model, and we performed the data visualization combined with ArcGIS.

## 4. Results and Discussion

### 4.1. Accessibility Results

The 15 min walking range of sites crawled by Python was used as the site range, and the site accessibility was measured with the area of the accessible range as an indicator. Overall accessibility shows a trend from high to low in the central region, while accessibility in the western end region is higher than in the east (Figure 3).

### 4.2. Analysis of the Spatial Distribution Characteristics of the Rail Transit Station Area Accessibility

#### 4.2.1. Analysis of the Rail Transit Station Area

Hotspot analysis of station area accessibility was conducted in ArcGIS. Figure 4 shows the spatial clustering of high accessibility and low accessibility with statistical significance, which more obviously shows the distribution characteristics of the high end and low end of the accessibility of Beijing rail transit stations. That is, the accessibility of rail transit stations within the fourth Ring Road is significantly higher than that in other areas, and there are two low accessibility clustering areas. One is the station area between the East fifth Ring Road and the East sixth Ring Road, and the other is the terminal station area of the Fangshan Line. This is highly similar to the construction time of the site and the surrounding site.

#### 4.2.2. The Spatial Clustering Characteristics of Beijing Rail Transit Station Area Accessibility

Figure 5 shows the spatial autocorrelation statistical results of the accessibility of Beijing rail transit stations. All the values pass the 1% correlation test and show significant clustering spatial clustering characteristics, indicating that the accessibility distribution of Beijing rail transit stations has a significant clustering phenomenon; that is, the accessibility of rail transit stations will affect the accessibility of surrounding stations.

Figure 6 shows the regional spatial autocorrelation results of the accessibility of Beijing rail transit stations. The distribution of the accessibility of Beijing rail transit stations presents an unbalanced state in space. There are five different cluster distribution regions in the featureless cluster, high-high cluster, high-low cluster, low-high cluster, and low-low cluster. On the whole, the inner part of the second ring is higher, the inner part between the second ring and the third ring is lower, and the northern part between the third ring and the fifth ring is higher than in other areas. Among them, the spatial clustering of the high-high cluster is more obvious. There are 56 stations with high accessibility, which affects the accessibility of surrounding stations, accounting for 12.84% of all stations, mainly distributed in the second ring road and the area between the North third Ring Road and the North fifth Ring Road.

### 4.3. Analysis of the Influencing Factors

#### 4.3.1. Accessibility Influencing Factors Mechanism Clustering Characteristics

Table 3 shows the results of the spatial autocorrelation analysis for the index data of each accessibility influence mechanism, and the results show that the selected data of 11 influencing factors have strong spatial autocorrelation. From the spatial autocorrelation index, the housing price, housing centrality, and public service density have strong clustering characteristics. However, the clustering characteristics of functional mixture degree and construction land proportion are the weakest. Z-scores describe how data values compare to the mean by indicating how many standard deviations a value falls above or below the mean. The *p* value represents significance.

#### 4.3.2. Random Forest Regression Analysis of the Factors Influencing Accessibility

In order to judge the influence degree of the selected index on the accessibility of the station domain, the SPSS Modeler 18.0 random forest algorithm was used to conduct regression analysis on the accessibility of the station domain and the selected 11 influencing factors.

To reduce the impact factors that influence each other between the deviation, the result is a random forest regression before the need to eliminate the multicollinearity of the relationship between impact factors. The impact factors need to be screened. The least square method (OLS) was used to analyze the site accessibility and 11 impact factors (Table 4). The Beta is the coefficient obtained after normalizing the data. The *p* value represents significance. VIF is collinear.

The results show the VIF (variance inflation factor) of functional density and cultural leisure density, indicating that there is a multicollinearity problem between these factors. The two indicators were excluded.

Random forest analysis was carried out on the remaining nine indicators. As shown in Figure 7, the fitting degree between the nine indicators and accessibility was 64%, indicating that the selected indicators had a strong explanatory ability for the accessibility of rail transit stations. The index significance results are shown in Figure 7, in which housing price, road density, distance from the center, and public service density can more significantly affect the station area accessibility. Residential centrality and functional mixing degree have significant effects on station accessibility. Commercial centrality density, population density, and construction land occupancy ratio have weak effects on station accessibility.

#### 4.3.3. Correlation Analysis of Factors Influencing Accessibility

In order to determine the influence direction of the selected influencing factors on the station accessibility, Pearson’s correlation analysis method was used to analyze the station accessibility and the nine selected influence indicators. The results are shown in Figure 8.

The results showed that housing price, public service density, residential centrality, functional mixing degree, commercial centrality, population density, and road density showed a significant positive correlation with station accessibility. There is a negative correlation between the distance from the center and the accessibility of the station area, indicating that the farther the distance from the center, the lower the station accessibility. However, the correlation between construction land proportion and site accessibility is not significant and fails the correlation test.

Based on the above correlation analysis and random forest regression analysis, according to the results, the 11 influencing factors were divided into significant promoting factors, general promoting factors, weak promoting factors, irrelevant factors, weak inhibiting factors, general inhibiting factors, and significant inhibiting factors to show the degree of influence of the selected influencing factors on the accessibility of the site area (Table 5).

### 4.4. Interpretation of the Results from MGWR

#### 4.4.1. Comparison of the Influence Mechanism Model

Through the above tests, the final remaining eight influence factors were used as variables to construct the site accessibility influence mechanism model construction. They are housing price, road density, public service density, functional mix, residential centrality, population density, commercial centrality, and distance from the center.

To select the best influence mechanism model, OLS, GWR, and MGWR were performed on the station accessibility and residual factors in turn, showing that the fitting R2 of the three models shows an increasing trend, and the AICc value shows a decreasing trend, and the difference is higher than three, indicating that the models have obvious differences [36] (Table 6).

Among the three models, the final R2 value of the MGWR model is 0.721, which is the group with the highest overall fitting degree, indicating that the accuracy of this model to explain the accessibility of Beijing rail transit stations is 72.1%. It can be concluded that the global accessibility model constructed by the OLS method does not consider the spatial difference characteristics of rail transit stations, and there are certain shortcomings. Although the GWR model considers the spatial difference characteristics of the sites, it adopts the unified bandwidth to analyze the impact factors and ignores the differences between different impact factors, which will cause some errors. This study shows that MGWR has certain advantages in the construction of the influencing mechanism model of rail transit station accessibility.

#### 4.4.2. Results Analysis of the MGWR Model

The statistical results of various coefficients of the MGWR model show (Table 7); the functional mixture degree, public service, housing price, residential center, and commercial center show positive effects on the accessibility of rail station areas. Population density has a negative effect. Distance from the center and road density presented positive and negative effects on language accessibility at different locations. Greater standard deviation indicates greater differences in the degree of accessibility in different rail transit station fields. The effect of the distance to the center, road density, and functional mixing degree on site accessibility differ more spatially than do the other influence factors.

### 4.5. Spatial Difference Analysis of Each Variable

The results are shown in Figure 9.

The functional mixing degree has the greatest impact on the accessibility of the rail transit station area with a positive impact, the higher the functional mixing degree of the rail transit station area, the higher the accessibility of the station area. In all rail transit stations, the influence of functional mixing degree on the accessibility of station area is relatively scattered. The positive relationship between this factor and station accessibility is mainly manifested in the station area between the East fourth Ring Road and the East fifth Ring Road, and between the West third Ring Road and the West fifth Ring Road. The influence of the functional mixing degree on the station heterogeneity between the fifth Ring Road and the sixth Ring Road also shows obvious spatial heterogeneity, and the accessibility of Beijing Subway line 6, Fangshan Line, and Daxing Line of Subway line 4 in the above two areas is more significant than that of other lines. The influence of the functional mixing degree inside the second ring on accessibility is lower than that outside the second Ring Road, which is due to the influence of policies in the second Ring Road and limited functional development, and the overall mixing degree is lower than that around the third Ring Road, so the influence is weak.The impact of the public service density on the station accessibility of rail transit stations has obvious spatial heterogeneity, and has a positive impact on the station accessibility; that is, the higher the public service density of rail transit stations, the higher the station accessibility. It shows a gradually significant influence from south to north; the influence of public service density is second only to the functional mixing degree. Spatially shows clear heterogeneity. The accessibility of rail transit stations located outside the North third Ring Road is higher than other stations affected by the density of public service. Among them, the public service density of Changping Line, the north section of Subway line 17, Subway line 15, and the end of Pinggu Line under construction have a strong positive impact on station accessibility. The influence of public service density on accessibility changes since the public service density itself shows a distinct hierarchical structure, and the central and southern parts are higher than the north, the impact of public service density on accessibility also shows this structure.The impact of the housing price on the station area accessibility has an obvious spatial heterogeneity; that is, the higher the housing price in the rail transit station area, the higher the accessibility in the station area. Stations with high housing prices indicate a higher level of development and better infrastructure, so accessibility is high.It has a positive impact on the station area accessibility. It shows a gradually significant trend of influence from west to east. The main affected stations are concentrated near the West sixth Ring Road, with the specific influence of Subway line 15, Subway line 6, Subway line 1/Batong Line and the eastern end of Pinggu Line under construction.The influence of residential centrality on station accessibility shows obvious spatial heterogeneity, and has a positive impact on station accessibility; that is, the closer the rail transit station is to the residential center, the higher the accessibility of the station is. It shows a gradually significant trend of influence from the southwest to northeast direction. Among them, it has the most significant impact on the northern terminal station area of Changping Line and the terminal station area of line 16.The influence of commercial centrality on the accessibility of the rail transit station shows obvious spatial heterogeneity, and has a positive impact on the station accessibility; that is, the closer the rail transit station is to the commercial center, the higher the station accessibility is. The overall performance is the trend of increasing influence from south to north, and the impact on the accessibility of the East fifth Ring Road and the North fifth Ring Road is higher than that in other regions. Among them, the accessibility of the northern terminal area of line 15, the northern terminal of Changping Line, and the station area of Pinggu Line is significantly affected by the commercial centrality.The influence of population density on the accessibility of station areas shows weak spatial heterogeneity and a negative influence on station accessibility; that is, the higher the population density within the range of rail transit stations, the lower the station accessibility. Population density has a significant impact on the accessibility of rail transit stations in the central region and the southwest region. The impact on the station area near and outside the sixth Ring Road is relatively low.The influence of the distance from the center on the station area accessibility shows the characteristics of polarization. The distance from the center has a positive impact on the accessibility of the station area outside the fifth Ring Road rail transit; that is, the farther the distance from the center, the higher the accessibility of the station area. However, the station domain accessibility has a negative impact on the station domain accessibility within the five rings; that is, the farther away from the center, the lower the station domain accessibility. The distance from the center has a significant impact on the accessibility of the external and central areas of the five rings. The level of development within the fifth ring road is relatively high, and the second ring road is the center to develop outward. The sub-center of the city and the Yizhuang Development Zone are located outside the fifth Ring Road, with a high level of infrastructure construction and high accessibility, so they are polarized.The influence of road density on the station accessibility overall shows a negative influence, with a positive influence on the accessibility of some stations near the sixth Ring Road of Fangshan Line and near the southeast corner of the fourth Ring Road. The station accessibility north of Chang’an Avenue and within the fourth Ring Road has a significant negative impact. This result is different from the finding that Dai et al. showed that road density is positively correlated with accessibility [28]. This is because this study considers the spatial heterogeneity of the influence of road density on accessibility.

The results of MGWR indicate the spatial heterogeneity of different factors. From the above eight factors affecting the accessibility of station area, five of the indicators have strong spatial heterogeneity, and each of the influence factors affects the degree and direction of the accessibility according to the different geographical locations. According to this result, when the government wants to improve the accessibility of the rail transit station area, it can accurately locate the priority factors for renovation and optimize them according to the spatial heterogeneity of different factors. For example, increasing the functional mix of stations between the East fourth Ring Road and the East fifth Ring Road can significantly improve the accessibility of the station area in the area. The improvement was more pronounced than in other regions.

## 5. Conclusions

In this study, the MGWR model was applied to the empirical study of RTX station area accessibility in Beijing. Taking the Beijing rail transit station area as the research object, 12 influencing factors affecting the vitality of neighborhoods were selected, and the spatial autocorrelation of each influencing factor and the accessibility of the rail transit station area were analyzed. By constructing the OLS model, GWR model, and MGWR model, eight influencing factors were finally selected. The spatial differentiation between the influence factors and the accessibility of rail transit stations was analyzed. The following conclusions are obtained:(1)The distribution of accessibility of the Beijing rail transit station area is unbalanced.
Overall, there is high accessibility within the fourth ring, and low accessibility in areas outside the fourth ring. In the outer region of the fourth Ring Road, the accessibility of rail transit stations distributed in the north–south direction is higher than that of the east–west direction; that is, the accessibility of stations near the North fifth Ring Road and the South fifth Ring Road is higher than that near the East fifth Ring Road.The station area within the fourth Ring Road is a hot-spot area, while the station area between the East fifth Ring Road and the East sixth Ring Road is a cold-spot area. Therefore, it is considered that the accessibility construction of the outer station area of the East fifth Ring Road should be strengthened.(2)Multiple factors lead to uneven distribution of rail transit station accessibility.
The factors influencing the accessibility of rail transit stations include road density, functional mixing degree, functional density, distance from the center, cultural and leisure density, public service density, population density, housing price, commercial centrality, and residential centrality. Population density, commercial centrality, and cultural leisure density have little influence on the accessibility of rail transit stations.Through the construction of the OLS regression model, GWR model, and MGWR model and comparison, the results show that the MGWR model has a higher degree of fitting with reality. Among them, the influence effects of five indicators are spatially heterogeneous, and in different geographical locations, the influence degree and direction on the accessibility of the station are different, and there are bidirectional effects. On the whole, there are five influencing factors of Beijing rail transit station accessibility with strong spatial heterogeneity.

This study verifies that the MGWR model can effectively explain the accessibility of the rail transit station area. The research methods of station area accessibility and the scope of application of MGWR have been broadened. Existing studies facilitate further in-depth study, but it must be acknowledged that there are certain limitations. Although the influence of spatial factors on accessibility in the rail transit station area is discussed through MGWR, the influence of changes in the surrounding environment of the rail transit station area and the interaction mechanism between spatial factors and accessibility have not been considered. Future research can focus on the relationship between the dynamic change mechanism around the station area and accessibility.

Improving the accessibility of rail transit stations is an important measure to provide high-quality travel services for commuters, and it is also an important way to improve the happiness of the last kilometer, which can reflect the high-quality development level of the city. The significance of this study is to describe the differences in accessibility of rail transit stations and study the related influencing factors. The significance of this study is to describe the differences in accessibility of rail transit stations and study their related influencing factors. This study has direct significance for guiding the development and optimization of the built environment around rail transit stations. With these findings, we can improve the quality of commuter travel and promote social equity through precise measures to improve the accessibility of rail transit stations.

## Figures and Tables

**Figure 1 ijerph-20-01535-f001:**
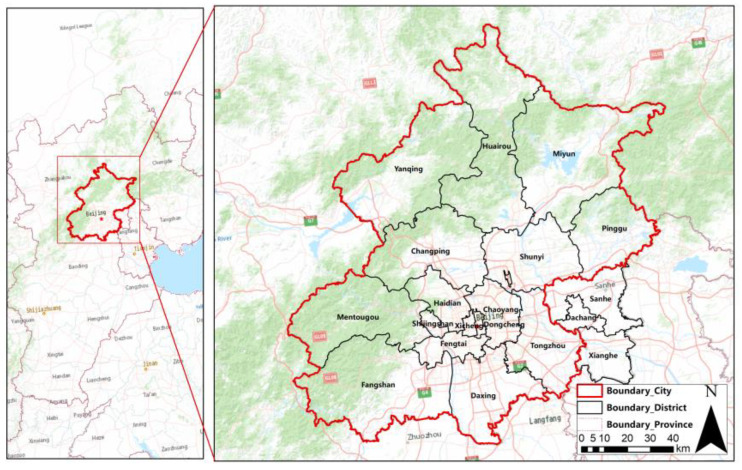
Study area.

**Figure 2 ijerph-20-01535-f002:**
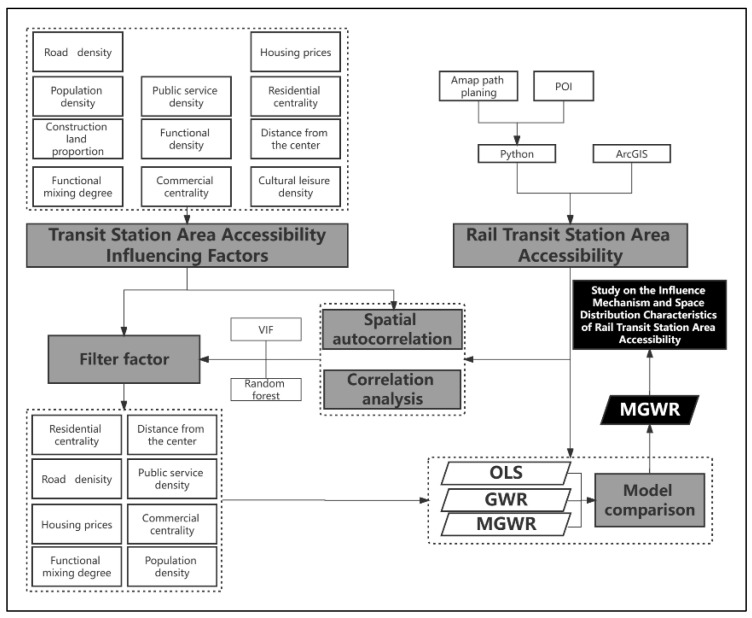
Research framework.

**Figure 3 ijerph-20-01535-f003:**
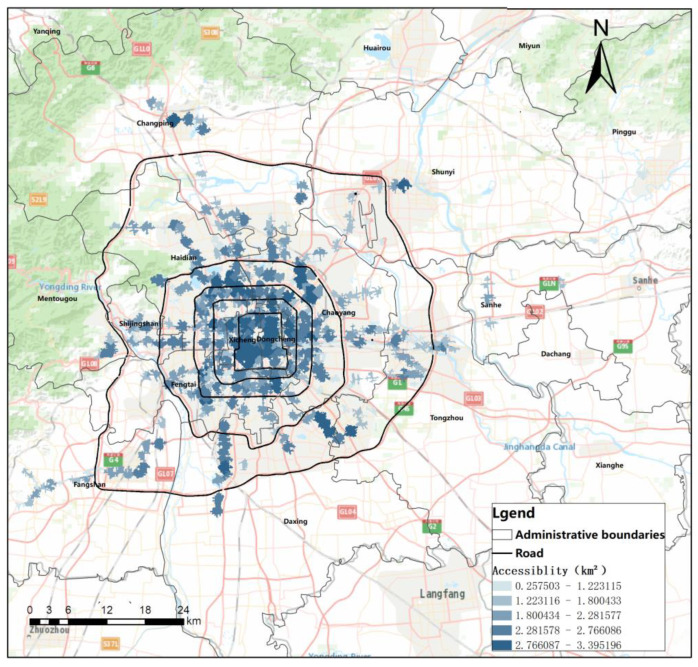
Distribution of rail transit station area accessibility in Beijing.

**Figure 4 ijerph-20-01535-f004:**
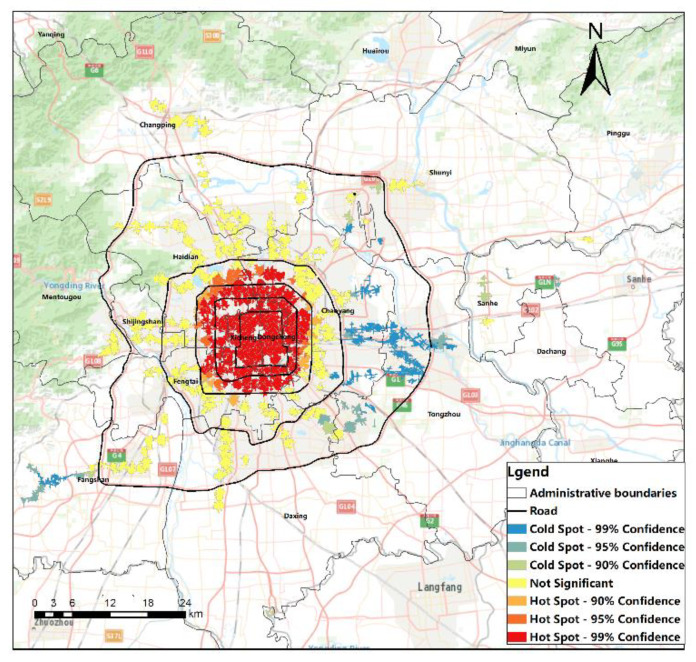
Hotspot analysis of Beijing rail transit station area accessibility.

**Figure 5 ijerph-20-01535-f005:**
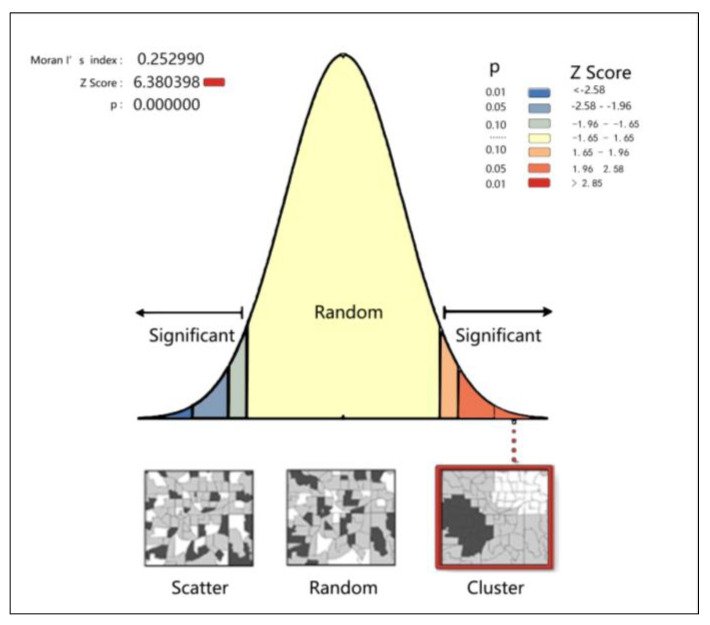
Spatial autocorrelation analysis of rail transit station area accessibility.

**Figure 6 ijerph-20-01535-f006:**
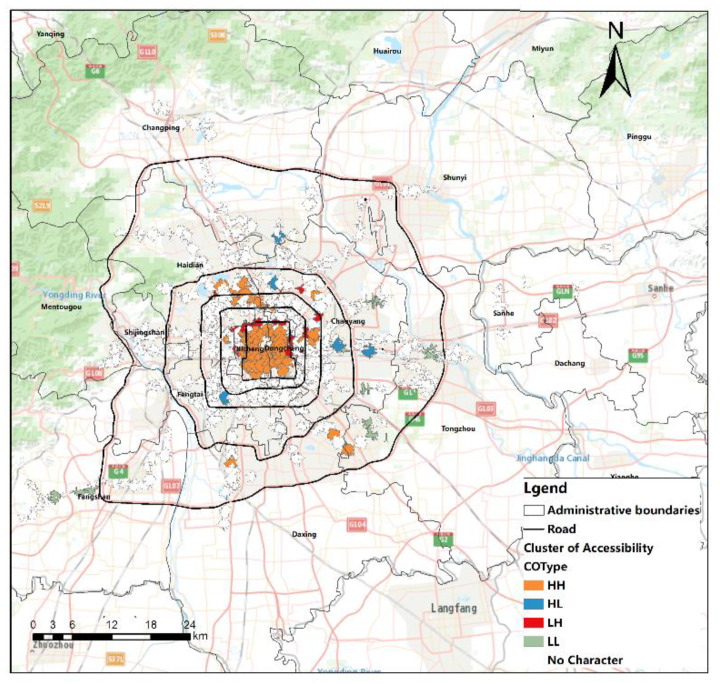
Regional spatial aggregation of rail transit station area accessibility in Beijing.

**Figure 7 ijerph-20-01535-f007:**
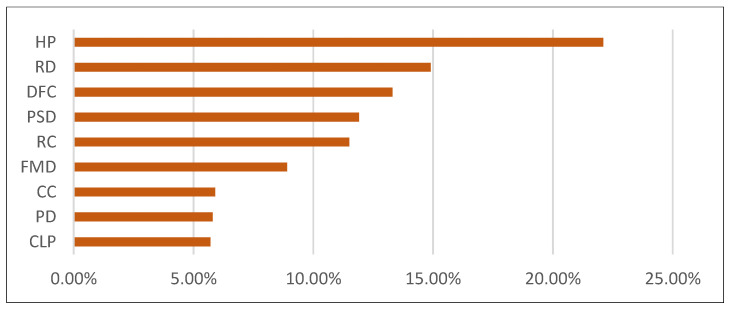
Significance of the influencing factor.

**Figure 8 ijerph-20-01535-f008:**
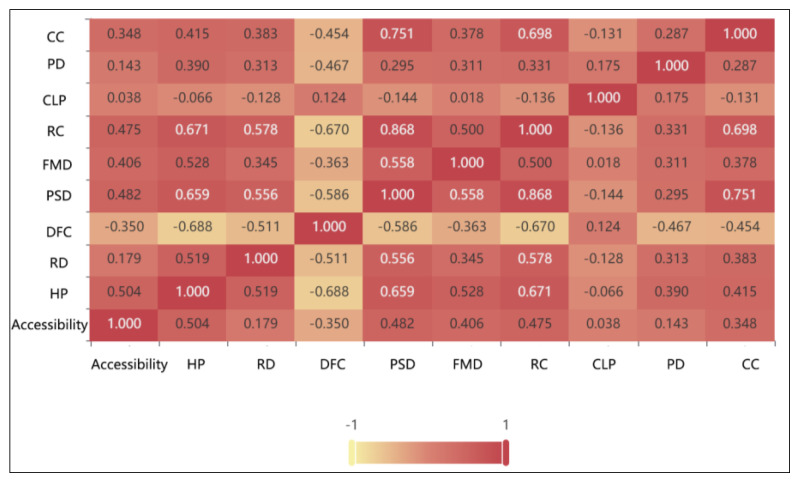
Correlation analysis matrix.

**Figure 9 ijerph-20-01535-f009:**
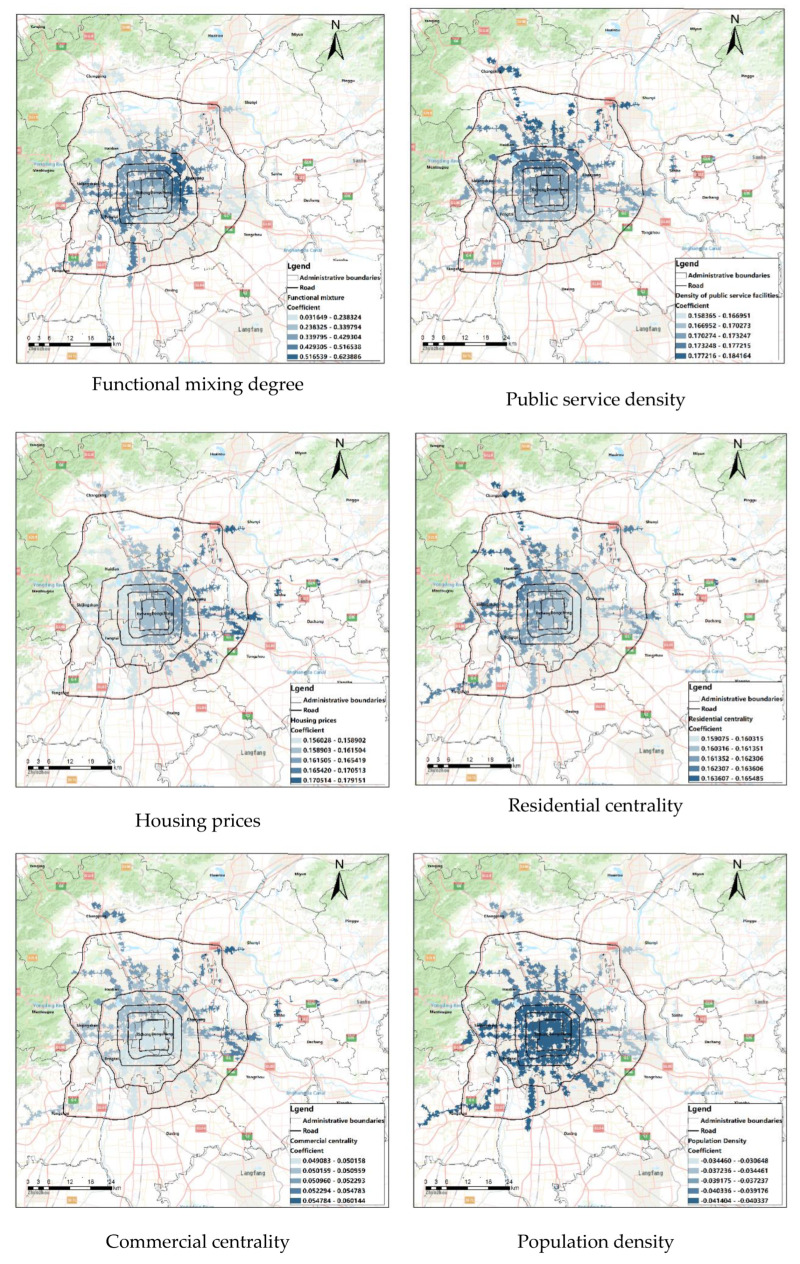
Spatial distribution of coefficients for MGWR.

**Table 1 ijerph-20-01535-t001:** Data source statistics.

Data Name	Obtained Time	Source	Formation
POI	July, 2022	Amap Open platform	Excel
Administrative boundaries	June, 2022	Aliyun Big Data platform	vector
Road network data	July, 2022	Amap Open platform	vector
Population distribution at the 100 m resolution	July, 2022	WorldPOP	grid
Housing estate price data	August, 2022	Lianjia	Excel
Construction land	August, 2020	GlobleLand30	vector

**Table 2 ijerph-20-01535-t002:** Calculation and statistics of the influencing factors.

Indicators		Calculation Method	Calculation Formula
First-Level Indicators	Second-Level Indicators		
Built environment	Road density (RD)	Ratio of road length to accessible area of each station area	$D_{ri}=\frac{L_{i}}{S_{i}}$ *D_ri_* represents the road density of the i-th site area; *L_i_* represents the total road length of the i-th station area; *S_i_* represents the accessible area of the i-th site domain
	Construction land proportion (CLP)	Ratio of construction land to accessible area of each station area	$D_{i}=\frac{M_{i}}{S_{i}}$ *D_i_* represents the construction land proportion of the i th site; *M_i_* represents the construction area of the *i* th site; *S_i_* represents the accessible area of the i-th site domain
	Functional mixing degree (FMD)	Shannon diversity values for the POI types in each site area	$H=-\Sigma_{i=1}^{k}p_{i}ln\left(p_{i}\right)$ *H* is the mixture of block functions, *i* is the number of POI categories within that block, *p_i_* is the ratio of the number of class *i* POI in the block to the total number of POI in the block
	Functional density (FD)	The ratio of POI quantity to accessible area	$D_{i}=\frac{M_{i}}{S_{i}}$ *D_i_* represents the POI function density of the i-th site domain; *P_i_* represents the number of POI types in the i-th site domain; *S_i_* represents the accessible area of the i-th site domain
	Distance from the center (DFC)	The straight-line distance from each station to Tian’anmen Square	
	Cultural leisure density (CLD)	Cultural and leisure facilities include sports and leisure, scenic spots, shopping services, catering service POI, the calculation formula is the same function density	Same as functional density
	Public service density (PSD)	Public service facilities include government agencies and social organizations, science, education and cultural services, health care services, and the POI of public facilities. The calculation formula is the same as the functional density	Same as functional density
Social environment	Population density (PD)	The ratio of population to accessible area in each station	$P_{i}=\frac{p_{i}}{S_{i}}$ *P_i_* represents the population density of the i-th site; *p_i_* represents the total population in the i th station; *S_i_* represents the accessible area of the i-th site domain
	Housing prices (HP)	Average housing price at each station area	$V_{i}=\frac{S_{i}}{N_{i}}$ *P_i_* represents the i-th station area housing price; *S_i_* represents the sum of the average house price of each residential area in the i-th station; *N_i_* represents the number of residential communities in the i-th site area
	Commercial centrality (CC)	Average nuclear density of commercial POI quantity, including shopping service, catering service, motorcycle service, car service, car maintenance, car sales POI	
	Residential centrality (RC)	Mean nuclear density of residential POI quantity, including business residential, residential service POI	

**Table 3 ijerph-20-01535-t003:** Statistics of spatial autocorrelation results of influencing mechanism of station area accessibility.

First-Level Indicators	Built Environment					Social Environment					
Second-Level Indicators	RD	CLP	FMD	FD	DFC	CLD	PSD	PD	HP	CC	RC
Moran’s I index	0.603446	0.261748	0.308113	0.630314	0.802362	0.481734	0.721675	0.460865	0.810325	0.508602	0.772155
Z scores	15.14644	6.616168	7.846358	15.83068	20.188620	12.140112	18.100903	11.571777	20.31261	12.821052	19.360691
*p* value	0.000 ***	0.000 ***	0.000 ***	0.000 ***	0.000 ***	0.000 ***	0.000 ***	0.000 ***	0.000 ***	0.000 ***	0.000 ***

***, **, * represent significance levels of 1%, 5%, and 10%, respectively.

**Table 4 ijerph-20-01535-t004:** Preliminary test results of VIF.

Indicators	PD	DFC	RD	HP	FMD	RC	CLD	FD	CC	PSD	CLP
Beta	−0.107	0.003	−0.197	00.363	0.136	0.197	0.093	0.019	−0.036	0.092	0.104
*p* value	0.024 **	0.964	0.000 ***	0.000 ***	00.007 ***	0.027 **	0.345	0.86	0.674	0.411	0.012 ***
VIF	1.474	2.561	1.719	2.751	1.645	5.167	6.428	7.853	4.938	8.251	1.129

***, **, * represent significance levels of 1%, 5%, and 10%, respectively.

**Table 5 ijerph-20-01535-t005:** Classification of the influencing factors.

Classification of the Correlation Results		Classification of the Random Forest Regression Results		
**Classification of positive and negative values**	**Promoting factors**	**Significantly promoting factors**	**General promoting factors**	**Weak promoting factors**
		Housing prices, road density	Public service density, functional mix degree, residential center	Population density, commercial centrality
	**Irrelevant factors**	The proportion of construction land		
	**Inhibiting** **factors**	Significant inhibiting factor	General inhibiting factor	Weak inhibiting factor
		Distance from the center		

**Table 6 ijerph-20-01535-t006:** Comparison of the model results.

Model Name	OLS	GWR	MGWR	
AICc	1049.035	1035.954	977.044	
R^2^	0.381	0.507	0.721	
Bandwidth		128	RD	46
			FMD	135
			DFC	43
			PSD	425
			PD	435
			HP	425
			CC	435
			RC	435

**Table 7 ijerph-20-01535-t007:** Statistics of the regression results of the MGWR model.

Influencing Factors	Average Value	Standard Deviation	Minimum	Median	Maximum
FMD	0.349	0.12	0.032	0.373	0.624
PSD	0.172	0.004	0.158	0.172	0.184
HP	0.162	0.004	0.156	0.16	0.179
RC	0.162	0.001	0.159	0.161	0.165
CC	0.051	0.002	0.049	0.05	0.06
PD	−0.04	0.002	−0.041	−0.04	−0.031
DFC	−0.315	0.45	−1.402	−0.433	1.073
RD	−0.374	0.25	−1.066	−0.37	0.108

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
