# Peer review of "Study on the Influence Mechanism and Space Distribution Characteristics of Rail Transit Station Area Accessibility Based on MGWR"

_ijerph, 2023, doi:10.3390/ijerph20021535_

Round 1

Reviewer 1 Report

Paper Review – International journal of Environmental Research and Public Health

Paper Title: “STUDY ON THE INFLUENCE MECHANISM AND SPACE DISTRIBUTION CHARACTERISTICS OF RAIL TRANSIT STATION AREA ACCESSIBILITY BASED ON MGWR”.

This paper aims to investigates the status quo and influencing 11 factors of the unbalanced distribution of rail transit station accessibility in Beijing. They applied three-models, respectively: (1) (Ordinary) Least Square (OLS) method (2), Geographical Weighted Regression (GWR) and Multi-scale Geographical Weighted Regression (MGWR). Moreover, they considers as influencing factors on rail transit station accessibility: Road density, Construction land proportion, Functional mixing degree, Functional density, Distance from the center, Cultural leisure density, Public service density, Population, Density, Housing prices, Commercial centrality, Residential centrality. The result of the research show spatial agglomeration characteristics in space and the influence of different influencing factors on the accessibility of the station area.

The article’s strength is the interesting approach in studying the spatial utilization pattern and influencing mechanism of rail station area

The article’s weakness consists of (1) some theoretical references absent and (correlated) clarity of the choices made; (2) lack of discussion on results, urban context, and previous research.

Here below are detailed suggestions to improve the quality and effectiveness of the paper:

Comment 1

Section 1 (Research background). There is a wide range of studies focusing on the spatial correlation between cities (urban context) and railway stations, which cannot be ignored. And with insights into accessibility that need to be implemented, especially in the European context that has been thinking about these issues for some time. Follow some references:

· Bertolini, L. (1999). Spatial development patterns and public transport: The application of an analytical model in the netherlands. Planning Practice and Research(14), 199-210. https://doi.org/10.1080/02697459915724.

· Carra, M. & Ventura, P. (2020). HSR stations’ urban redevelopments as an impulse for pedestrian mobility. An evaluation model for a comparative perspective. In M. Tira, M. Pezzagno & A. Richiedei (a cura di), Pedestrians, Urban Spaces and Health. CRC Press, London, pp. 120-124. https://doi.org/10.1201/9781003027379.

· Vale, D. S. (2015). Transit-oriented development, integration of land use and transport, and pedestrian accessibility: Combining node-place model with pedestrian shed ratio to evaluate and classify station areas in Lisbon. Journal of Transport Geography, 45, 70-80. https://doi.org/10.1016/j.jtrangeo.2015.04.009.

· Bertolini, L. (1996). Nodes and places: complexities of railway station redevelopment. European Planning Studies, 4(3), 331-345.

· Bowes, D. R. (2001). Identifying the Impacts of Rail Transit Stations on Residential Property Values. Journal of Urban Economics(50), 1-50.

· Caset, F., Vale, D. S., & Viana, C. M. (2018). Measuring the Accessibility of Railway Stations in the Brussels Regional Express Network: a Node-Place Modeling Approach. Networks and Spatial Economics, 18(3), 495-530.

Comment 2

Section 1 (Research background). Based on the above, we found that past studies are limited to the following aspects: Research on rail transit accessibility rarely focuses on the level of station area, which is difficult to reflect the service level of the last kilometer of rail transit travel”. This is not true. For example, in the previous references. Other reference considers station area accessibility applying a backtracking algorithm that optimises distances (accessibility) by determining the pedestrian path with the shortest distance:

·         Carra, M., Rossetti, S., Tiboni, M., & Vetturi, D. (2022). Urban Regeneration Effects on Walkability Scenarios. An application of space-time assessment for the people-and-climate oriented perspective. TeMA Journal of Land Use, Mobility and Environment, 101-114. https://doi.org/10.6093/1970-9870/8644

Comment 3

Section 1 (Research framework). A summary of the structure of the article is missing. It can be linked to the parts written in the section.

Comment 4

Section 1 (Research framework) and Section 2. The Figure 1 and Section 2 are specific of method application to the case study, so I would move it after the methodological Section (3).

Comment 5

Taking the current status of rail transit stations in Beijing as an example, the calculated time cost is 15 minutes, and the range within a 15-minute walk around each station is analyzed respectively”. However, I could not understand what speed was adopted in this case. Speed is indeed a very important parameter that directly affects accessibility (as previous references show).

Moreover, does “time cost” take into account any slowdowns? For instance, the following reference considers a waiting time at intersections equal to 0.4 to unsignalised pedestrian crossing and 0.8 to signalised ones.

·         Caselli, B., Carra, M., Rossetti, S., & Zazzi, M. (2021). From urban planning techniques to15-minute neighbourhoods. A theoretical framework and GIS-based analysis of pedestrian accessibility to public services. European Transport/Trasporti Europei, 85, pp. 1-15. https://doi.org/10.48295/ET.2021.85.10

Comment 6

In section 3, the authors say: According to previous studies, the accessibility of rail transit station area is related to a series of factors, including road density, population density, development degree, land use nature, etc”. I have some concerns about factors. For example, the same architectural typology of the station and surrounding conditions determine its accessibility and so on. This can be implemented by other studies focused on the topic. Leaving aside this aspect which would distort the whole work (but which can be pointed out in the text), the road density indicator used raise my doubts. If the analysis concerns pedestrian accessibility to stations, how could a generic road density have been used? Is it considered equivalent to pedestrian paths? Does it take into account streets where pedestrian mobility is denied (e.g., highway) or paths entirely dedicated to pedestrians (paths in greenery)?

One more clarification., the functional density is very similar to cultural leisure density, Public service density, and Commercial centrality. Why were POIs considered both at an aggregated and disaggregated level?

Comment 7

In section 4, the authors say: “The negative correlation with center distance and station accessibility indicates that the lower the station accessibility is from the center.”. I think the sentence is misspelled.

Comment 8

Section 4.5. Spatial Difference Analysis of Each Variable. The section is very interesting and presents a correct reading of the results. However, the reason is not explained in reality, in other words there is no discussion of the results either on the urban context or to previous literature. I think this would make the authors' contribution even better and stronger.

Comment 9

Section 1 (Research background). The authors say: “It does not take 121 into account the spatial difference between the built environment and the social environ-122 ment of the station space on the impact of accessibility.True and interesting. However, no consideration is made of this in the results or conclusions.

Author Response

Please see the attachment:

Reviewer 2 Report

1.        A clear and urgent practical problem should be given directly in the introduction, stating the purpose and significance of the study.

2.        Section 1.2 should appear in the methods section, not in the introduction.

3.        In the determination of the station area a radius of 15 minutes on foot is used, what is the absolute value of the specific spatial distance in meters, which should be stated in 2.2.2.

4.        The study population was selected to span the administrative regions outside of Beijing and the specific location of the study sites should be illustrated, rather than just a macro map of the study area.

5.        8 out of 11 spatial factors affecting accessibility were finally selected for this study. The selection conditions and criteria are not explicitly stated in the article and are recommended to be stated before section 4.4.1.

6.        The conclusion should compare the strengths or weaknesses of the study in the context of previous research results and further indicate the direction of further research.

7.        This study focuses on the basic research on the mechanism of environmental factors affecting accessibility in the station area, and the article should further point out the relevant strategies to optimize the space of Beijing metro station area with the aim of improving accessibility based on the above-mentioned results, which is currently too weak.

8.        The tense of the article should be corrected, e.g., in the abstract the findings and process should be in the past tense.

Author Response

Please see the attachment:

Reviewer 3 Report

Review Report by Reviewer 3

This article “Study on the Influence Mechanism and Space Distribution Characteristics of Rail Transit Station Area Accessibility Based on MGWR. This study used Multisource data and applied two spatially sound model such as GWR and MGWR. This study can made sufficient contributions to the existing literature. I have few concerns that needs to address to improve the quality of manuscript.

Abstract:  The abstract needs to be concise and make 300 words by keeping the basic structure (IMRAD).

Introduction

1.      Line no 43 needs be revised change the continuous to gathers

2.     Change the zuo  capital words into small in line no 65.

3.     Line 70 and 88  need to be properly cited  kusmo and shen.

4.     Clarify the contribution of study, change the subsection 1.2 into framework and contribution or objective of the study to clearly convey the intending meaning to reader.

5.     Figure 1 has spelling mistakes

6.     It would be better to provide accessibility different types of accessibility in literature such as locational accessibility, utility based, distance based, individual based etc and why choose your distance-based accessibility. This paper could help you the latest literature Analyzing Spatial Location Preference of Urban Activities with Mode-Dependent Accessibility Using Integrated Land Use- Transport Models

7.     Change the research object 2.1 into study area.

8.     Separate the scale and arrow from the legend. Move the arrow to the top right side and scale it bottom left side.

9.     Properly cite the link in references line 164,177, and 182.

10.  Remove “in” an keep only July from table source statistics and keep the words capital order correct in entire table.

11.  Rephrase line 197 and 279.

12.  Write proper name IN TABLE 3 like Z value and p value the p value should be 0.0001 or 0.01 or 0.001 and be correct it . to show it with which significant level. 5% ,10% or 1% or 0.1%

13.  Similarly for table 4 clarify the b is beta ( or coefficient )  p value and VIF.

14.  It would be nice to compare with your results with previous studies discussed int eh literature.

15.  The conclusion is well written.

Thanks 

Kindly see the attached file.

Round 2

Reviewer 1 Report

I renew my previous review as it is impossible for me to verify from the uploaded file what has been modified (in fact the text is difficult to understand due to the formatting revisions, and I don't know why but I note some comments from the reviewers of the text).

Therefore, I ask the authors to upload the clean file with the changes marked in red and a second response file to the reviewers' comments with the changes/reasons exposed.

Author Response

Please see the attachment:

Reviewer 2 Report

  • The authors have revised and improved the opinions and suggest publication.

Reviewer 3 Report

Thanks to the authors for their responses but still there is some issues exist.

--Paper English writing style needs to be improved. In the abstract second sentence has an issue. Need to properly revise it whole paper.

--line 139-142 needs to revise it. it's unclear to the reader

--- Don't use geographically weighted regression everywhere. To make it concise and readable use it in the introduction with an abbreviation then use its abbreviation in subsequent sections.

---where is the reference number of cao etl and zhang et al in lines 179 and 191?

--- introduction part contains, background and worldwide scope, a summary of the literature, significance and gaps, and how you are going to fill the gap. the last point how are you going to fill the gaps is still missing. after line215  there should be an objective and closing paragraph in the introduction that this study is going to bridge these gaps by using this method and state their significance which is still unclear to me. if you don't understand then see these papers and get ideas from them. Evaluating Locational Preference of Urban Activities with the Time-Dependent Accessibility Using Integrated Spatial Economic Models, Analysis of the Influential Factors towards Adoption of Car-Sharing: A Case Study of a Megacity in a Developing Country. write it clearly your paragraph is not clear.

---- you have explained the framework in the method section but your contribution must be a concise paragraph of the introduction.

---Z-scores describe how data values compare to the mean by indicating how many standard deviations a value falls above or below the mean. Rewrite it

---use the super script of R2

--- You have not compared the results with previous studies see this paper which is similar to your study and contains the latest literature that will help you. for example in the housing price and city center accessibility. Exploring the Effects of Transportation Supply on Mixed Land Use at the Parcel Level

Thanks

Round 3

Reviewer 1 Report

Thanks to the authors for their responses. However, there are still minor tweaks to fine-tune. In particular, it concerns many English writing style errors and typos.

For instance, in section 1, it concerns rows: 9, 29, 52, 54, 70, 90, 92, 94, 103, 106, etc.

The authors should carefully review and correct all text of the paper.